# Temporal Abstractions-Augmented Temporally Contrastive Learning: An Alternative to the Laplacian in RL

**Akram Erraqabi**[1,2]     **Marlos C. Machado**[4,5,6]     **Mingde Zhao**[1,3]     **Sainbayar Sukhbaatar**[8]

**Alessandro Lazaric**[8]     **Ludovic Denoyer**[8]     **Yoshua Bengio**[1,2,7]

[1]Mila   [2]Université de Montréal   [3]McGill University   [4]Amii
[5]University of Alberta   [6]CIFAR AI Chair   [7]CIFAR Fellow   [8]Meta AI

## Abstract

In reinforcement learning, the graph Laplacian has proved to be a valuable tool in the task-agnostic setting, with applications ranging from skill discovery to reward shaping. Recently, learning the Laplacian representation has been framed as the optimization of a temporally-contrastive objective to overcome its computational limitations in large (or continuous) state spaces. However, this approach requires uniform access to all states in the state space, overlooking the exploration problem that emerges during the representation learning process. In this work, we propose an alternative method that is able to recover, *in a non-uniform-prior setting*, the expressiveness and the desired properties of the Laplacian representation. We do so by combining the representation learning with a skill-based covering policy, which provides a better training distribution to extend and refine the representation. We also show that a simple augmentation of the representation objective with the learned temporal abstractions improves dynamics-awareness and helps exploration. We find that our method succeeds as an alternative to the Laplacian in the non-uniform setting and scales to challenging continuous control environments. Finally, even if our method is not optimized for skill discovery, the learned skills can successfully solve difficult continuous navigation tasks with sparse rewards, where standard skill discovery approaches are no so effective.

## 1 INTRODUCTION

With the advent of deep reinforcement learning [Mnih et al., 2015], representation learning [c.f. Bengio et al., 2013] has become one of the main topics of interest in reinforcement learning (RL). In fact, learning in environments with rich

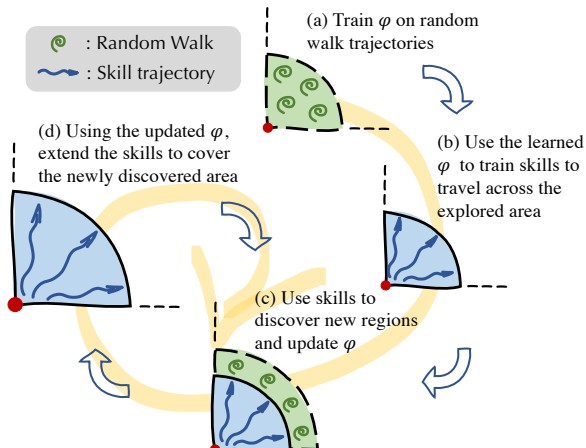

Figure 1: Our representation is trained to encode the area that the agent has learned to cover. Skills are continuously trained on the representation to discover new areas where novel data is collected to refine the representation, progressively extending its coverage. Similar incremental discovery is at the core of other works [Ecoffet et al., 2021, Pong et al., 2020, Machado, 2019].

observations and complex dynamics [e.g., Bellemare et al., 2020] has motivated recent works on learning representations, e.g. controllable or contingent features [Bengio et al., 2017, Choi et al., 2019] on top of which one can potentially learn latent models in the perspective of planning [Hafner et al., 2019, Nasiriany et al., 2019, Schrittwieser et al., 2020] and control [Watter et al., 2015, Banijamali et al., 2018, Hafner et al., 2020].

In this work, we are interested in the task-agnostic setting in which an RL agent first interacts with the environment to build a representation, $\phi$, of the state space, $\mathcal{S}$, without relying on any specific reward signal. This representation can later be used to solve a task posed in the environment in the form of a reward function. In this setting, the environment dynamics are the main informative interaction

*Accepted for the 38th Conference on Uncertainty in Artificial Intelligence* (UAI 2022).

channel available to the agent. This has motivated the use of graph Laplacian-based methods where the graph vertices correspond to the states and its edges to the transitions probabilities. The Laplacian's eigenvectors can been leveraged as a holistic state representation, termed the Laplacian representation, which captures the environment dynamics [Mahadevan, 2005, Mahadevan and Maggioni, 2007].

Wu et al. [2019] have recently proposed an efficient approximation of the Laplacian representation (LAP-REP) by framing the graph drawing objective as a temporally-contrastive loss (see Section 2.2). This formulation works around potentially prohibitive eigendecompositions and extends the Laplacian's applicability to large (and continuous) state spaces. However, it assumes access to a uniform sampling prior over all states in the state space. In practice, this translates in the ability to reset the agent to any state in the environment, which artificially alleviates the exploration problem. As we show in Section 5, this assumption is crucial for the quality of the learned representation. In the absence of the uniform prior privilege, such sampling is not trivial to achieve since the agent has to first explore and learn about the state space to be able to access arbitrary states. In effect, one must handle the exploration along the representation learning in order to preserve the quality of the representation. In this work, we propose TATC, an alternative representation learning framework to LAP-REP that extends a similar temporally-contrastive approach to a non-uniform-prior setting while preserving the desired properties and quality.

In practice, the representation is trained on data collected with a uniformly random policy. However, without a uniform access to the state space, the collected data is concentrated around accessible starting states. To achieve better data collection, we tie the representation learning problem to that of learning a covering strategy. Briefly, our method consists in using the available representation to train a skill-based covering policy that is in turn used to discover yet unseen parts of the state space, providing novel data to refine and expand the representation. Our approach, illustrated in Figure 1 shares a similar motivation with several previous works [Machado, 2019, Machado et al., 2017, 2018, Jinnai et al., 2020]. In addition to the aforementioned virtuous learning cycle between representation and skills, we propose to integrate the temporal abstractions captured by the skills in the contrastive representation learning objective. This augmentation contributes to a better temporally-extended exploration and enforce the representation's *dynamics-awareness*, i.e. how representative the representation-induced metric is of distances in the state space.

We empirically show our agent's ability to progressively explore the state space and consistently extend the domain covered by representation in a non-uniform-prior setting. We show that our representation leads to better value predictions than LAP-REP, and that it recovers the representation

quality expected from a uniform prior. We also evaluate our representation in shaping rewards for goal-achieving tasks, and we show it outperforms LAP-REP, confirming both its superior ability in capturing dynamics and in scaling to *larger* environments. Finally, the skills learned with our framework also prove to be successful at difficult continuous navigation tasks with sparse rewards, where other standard skill discovery methods have limited efficacy.

## 2 PRELIMINARIES

### 2.1 TASK-AGNOSTIC RL

We describe a task-agnostic reinforcement learning (RL) environment as a task-agnostic Markov decision process (MDP) $\mathcal{M} = (\mathcal{S}, \mathcal{A}, P, \gamma, d_0)$ where $\mathcal{S}$ is the state space, $\mathcal{A}$ the action space, $P : \mathcal{S} \times \mathcal{A} \to \Delta(\mathcal{S})$ is the transition dynamics defining the next state distribution given the current state and action taken, $\gamma \in [0, 1)$ the discount factor, and $d_0$ is the initial state distribution. A policy $\pi : \mathcal{S} \to \Delta(\mathcal{A})$ maps states $s \in \mathcal{S}$ to distributions over actions. We denote by $\Delta$ the probability simplex.

Knowledge acquired from task-agnostic interactions with the environment (e.g., a representation or a policy) can then be leveraged for specific tasks. A task is instantiated with a reward function, $R : \mathcal{S} \to \mathbb{R}$, which is combined with the task-agnostic MDP. The task objective is to find the optimal policy maximizing the expected discounted return, $\mathbb{E}_{\pi, d_0}\left[\sum_t \gamma^t R(s_t, a_t)\right]$, starting from state $s_0 \sim d_0$ and acting according to $a_t \sim \pi(\cdot|s_t)$.

### 2.2 THE LAPLACIAN REPRESENTATION

The Laplacian representation (LAP-REP), as proposed by Wu et al. [2019], can be learned with the following contrastive objective:

$$\mathcal{L}_{Lap}(\phi; \mathcal{D}_{\pi_\mu}) = \mathbb{E}_{(u,v) \sim \mathcal{D}_{\pi_\mu}}\left[\|\phi(u) - \phi(v)\|_2^2\right] +$$
$$\beta \; \mathbb{E}_{\substack{u \sim \mathcal{D}_{\pi_\mu} \\ v \sim \mathcal{D}_{\pi_\mu}}}\left[(\phi(u)^\top \phi(v))^2 - \|\phi(u)\|_2^2 - \|\phi(v)\|_2^2\right], \quad (1)$$

where $\beta$ is a hyperparameter, $\phi : \mathcal{S} \to \mathbb{R}^d$ is a $d$-dimensional representation, $\pi_\mu$ the uniformly random policy (random walk trajectories), $\mathcal{D}_{\pi_\mu}$ a set of trajectories from $\pi_\mu$ (random walks). We use $(u, v) \sim \mathcal{D}_{\pi_\mu}$ to denote the sampling of a random transition from $\mathcal{D}_{\pi_\mu}$, and similarly $u \sim \mathcal{D}_{\pi_\mu}$ for a random state. Wu et al. [2019] showed the competitiveness of the Laplacian representation when provided with a uniform prior over $\mathcal{S}$ during the collection of $\mathcal{D}_{\pi_\mu}$. This objective is a *temporally-contrastive* loss: it is comprised of an attractive term that forces temporally close states to have similar representations, and of a repulsive term that keeps far apart temporally far states' representations.

Here, the repulsive term was specifically derived from the orthonormality constraint of the Laplacian eigenvectors.

## 2.3 THE NON-UNIFORM PRIOR SETTING

In RL, representation learning is deeply coupled to the problem of exploration. Indeed, the induced state distribution defines the representation's training distribution. For instance, LAP-REP [Wu et al., 2019] has been learned in the specific *uniform prior*. In this setting, $\mathcal{D}_{\pi_\mu}$, from Equation (1), is a collection of trajectories with uniformly random starting states, which provides a uniform training distribution to the representation learning objective. In the case of a non-uniform prior, the induced visitation distribution can be quite concentrated around the starting states distribution when solely relying on random walks to collect data, hence the need for a better exploration strategy in order to achieve a better training distribution for $\phi$.

To study the problem described above, we investigate the setting in which the environment has a fixed predefined state $s_0$ to which it resets with a probability $p_r$ every $K$ steps; with $K$ of the order of diameter of $\mathcal{S}$. With a uniformly random behavior policy, this setting is equivalent to a initial state distribution that is concentrated around $s_0$ and whose density decays exponentially away from it. We will refer to this setting as the *non-uniform-prior* (non-$\mu$) setting, as opposed to the *uniform-prior* ($\mu$) setting where the agent has uniformly access to the state space.

# 3 TEMPORAL ABSTRACTIONS AUGMENTED REPRESENTATION LEARNING

In this section, we present Temporal Abstractions-augmented Temporally-Contrastive learning (TATC), a representation learning approach in which the representation works in tandem with a skill-based covering policy for a better representation learning in the non-uniform prior setting.

Before presenting the components of TATC and how they are trained, we first provide an intuitive description of how it operates in the non-uniform prior setting. A pseudo-code of the algorithm can be found at the end of this section.

## 3.1 TATC: A SKETCH OF THE ALGORITHM

In the non-uniform-prior setting, the agent is reset to a fixed state $s_0$ after each episode with some probability $p_r$. At the beginning of each episode (not necessarily at $s_0$, due to the probabilistic resetting), our agent can choose (with some probability $p_{rw}$) either to follow the uniform policy $\pi_\mu$ or to act according to an exploratory skill-based policy. Random walk data are, similarly to Wu et al. [2019], collected to train the representation, while the skills are trained to extend the

area of $\mathcal{S}$ that the agent is able to efficiently reach. In order to leverage the rich compositionality of skills, the agent executes a sequence of $L$ consecutive skills each time it decides to call the skill-based policy. Initially, trajectories from $\pi_\mu$ cover the vicinity of $s_0$, making the representation reliable in that area, i.e. representative of its dynamics. Inevitably, the skills trained on this representation benefit from its emerging structure and progressively gain *behavioral* structure: they allow the agent to travel efficiently across this explored area. In other words, the agent becomes capable of reaching the frontier of the explored regions faster, and it is able to collect, using $\pi_\mu$, novel data for the representation. The latter is hence refined, and its coverage extended. With a refined representation, the skills are able to reach even further areas. This process emerges as a virtuous collaboration between the representation and the skills, where both benefit from improving each other. Eventually, by acquiring more knowledge from unexplored areas, the agent helps overcoming the loss in the representation's expressiveness that we observed when solely relying on $\pi_\mu$ in the non-uniform-prior setting.

In the remainder of this section we first propose a generic alternative objective to Equation (1) that suits the non-uniform prior setting, then we describe the exploratory policy training. Finally, we introduce an augmentation of the proposed objective based on the learned temporal abstractions, to improve exploration and enforce the dynamics-awareness of the representation.

## 3.2 TEMPORALLY-CONTRASTIVE REPRESENTATION OBJECTIVE

As mentioned in Section 2.2, the repulsive term in LAP-REP's objective, in Equation (1), derives from the eigenvectors' orthonormality constraint. However, because the environment is expected to be progressively covered in the non-uniform prior setting, the orthonormality constraint can make online representation learning highly non-stationary.[1] For this reason, we adopt the following objective with a generic repulsive term that is more amenable to online learning:

$$\mathcal{L}_{cont}(\phi; \mathcal{D}_{\pi_\mu}) \triangleq \mathbb{E}_{(u,v)\sim\mathcal{D}_{\pi_\mu}} \left[ \|\phi(u) - \phi(v)\|_2^2 \right] + \beta \; \mathbb{E}_{\substack{u\sim\mathcal{D}_{\pi_\mu} \\ v\sim\mathcal{D}_{\pi_\mu}}} \left[ \exp(-\|\phi(u) - \phi(v)\|_2) \right]. \quad (2)$$

In the following, we describe our representation-based skills training framework. Beyond addressing the exploration need, these skills will later (Section 3.4) be used to augment the objective above to obtain TATC's representation learning objective.

---

[1]In general, even within a given matrix's perturbation neighborhood, its eigenvectors can show a highly nonlinear sensitivity [Trefethen and Bau, 1997].

### 3.3 REPRESENTATION-BASED SKILLS TRAINING

In the non-uniform-prior setting, exploration is required to provide the representation with a better training distribution. To this purpose, we adopt a hierarchical RL approach to leverage the exploration efficiency of skills, also known as options [Sutton et al., 1999]. Let $\phi : \mathcal{S} \to \mathbb{R}^d$ be our $d$-dimensional representation. The agent acts according to a bi-level policy $(\pi_{\text{hi}}, \pi_{\text{low}})$. The high-level policy $\pi_{\text{hi}} : \mathcal{S} \to \Delta(\Omega)$ defines, at each state $s$, a distribution over a set $\Omega$ of directions (unit vectors) in the representation space ($\Omega = \{\boldsymbol{\delta} \mid \boldsymbol{\delta} \in \mathbb{R}^d, \|\boldsymbol{\delta}\|_2 = 1\}$). Each direction corresponds to a fixed length skill encoded by the low-level policy $\pi_{\text{low}} : \mathcal{S} \times \Omega \to \Delta(\mathcal{A})$. These skills are trained to travel *in the representation space* along the directions instructed by $\pi_{\text{hi}}$. In short, given a sampled direction $\pi_{\text{hi}}(\cdot|s) \sim \boldsymbol{\delta} \in \Omega$, the low-level policy executes the skill $\pi_{\text{low}}(\cdot|s, \boldsymbol{\delta})$ for a fixed number of steps $c$ before $\pi_{\text{hi}}$ is called again.

Now, we describe the intrinsic rewards used to train the policies $\pi_{\text{low}}$ and $\pi_{\text{hi}}$.

**Low-level Policy.** $\pi_{\text{low}}$ is trained to follow directions picked by $\pi_{\text{hi}}$ in the representation space. For a given $\boldsymbol{\delta} \in \Omega \subset \mathbb{R}^d$, the corresponding skill $\pi_{\text{low}}(\cdot|s, \boldsymbol{\delta})$ is trained to maximize the reward function:

$$r^{\boldsymbol{\delta}}(s, s') \triangleq \frac{(\phi(s') - \phi(s))^\top \boldsymbol{\delta}}{\|\phi(s') - \phi(s)\|_2} \quad , \qquad (3)$$

where $(s, s')$ is an observed state transition, and $\phi$ the representation being learned. We use the cosine similarity as a way to encourage learning diverse directional skills. Indeed, skills co-specialization is avoided by rewarding the agent for the steps induced along the instructed direction $\boldsymbol{\delta}$ regardless of their magnitudes.

**Connection to Behavioral Mutual Information.** It is worth noting that our reward design can be interpreted as a mutual-information-based intrinsic control. We provide more details on this connection in Appendix E.

**High-level Policy.** The high-level policy guides the covering strategy. It does so by sampling the skills of the most promising directions in terms of the amount of exploration, measured by the travelled distance in the representation space. For this purpose, we design a reward function defined over a sequence of $L$ consecutive skills. Let $\{s_k^{\text{hi}}\}_{k=1}^{L}$ be the sequence of their initial states and their respective sampled directions, $\boldsymbol{\delta}_k \sim \pi_{\text{hi}}(\cdot|s_k^{\text{hi}})$. Since $\phi$ is trained to capture the dynamics, the travelled distance in $\phi$'s space is a proxy of how far the agent has moved along the MDP dynamics. In other words, the dynamics-awareness property provides a convenient way to quantify how far the choices made by $\pi_{\text{hi}}$ eventually brought the agent, which can be used to evaluate their exploratory potential. This observation has

motivated the proposed form of the high-level reward which we describe in the following. Given a high-level trajectory, $\tau^{\text{hi}} = (s_1^{\text{hi}}, s_2^{\text{hi}}, ..., s_L^{\text{hi}}, s_f^{\text{hi}})$, with $s_f^{\text{hi}}$ the final state reached by the last skill, the high-level policy is trained to maximize the quantity:

$$\forall k \in \{1, ..., L\}, R^{\text{hi}}(s_k^{\text{hi}}, \boldsymbol{\delta}_k) \triangleq \|\phi(s_1^{\text{hi}}) - \phi(s_f^{\text{hi}})\|_2 \quad , \quad (4)$$

where $\boldsymbol{\delta}_k \sim \pi_{\text{hi}}(\cdot|s_k^{\text{hi}})$ is the direction sampled at $s_k^{\text{hi}}$. From the policy optimization perspective, each of these quantities plays the role of the return cumulated along the sampled high-level trajectory and *not* just a single (high-level) step reward. This term looks at reaching $s_f^{\text{hi}}$ as the result of a sequential collaboration of $L$ skills, rewarding them equally. It values how far the whole sequence of skills has eventually travelled which is the result of selecting and executing each of the sampled directions $\boldsymbol{\delta}_k$ within the sequence. Thus, this term does depend directly, *even if not explicitly*, on those directions. Finally, we provide a discussion on the advantage of our high-level reward design (4) over a greedy one in Appendix B.

In the following section, we show how the skills trajectories can be used to augment the representation with the learned temporal abstractions

### 3.4 AUGMENTING REPRESENTATION LEARNING WITH TEMPORAL ABSTRACTIONS

A skill abstracts a temporally-extended behavior in a single (high-level) action. As a *temporal abstraction*, it represents a factorized knowledge of the environment dynamics in the form of a policy. Here, we propose to integrate these temporal abstractions to the representation objective to better capture the environment dynamics. In order to preserve the temporal contrast of the base objective (2), we augment it with the following contracting term along skills trajectories:

$$\mathcal{B}(\phi; \mathcal{D}_s) \triangleq \mathop{\mathbb{E}}_{\substack{\tau_{\boldsymbol{\delta}} \sim \mathcal{D}_s \\ \tau_{\boldsymbol{\delta}} = (s_0, ..., s_c)}} \left[ \sum_{k=0}^{c-1} \|\phi(s_k) - \phi(s_{k+1})\|_2 \right], \quad (5)$$

where $\mathcal{D}_s$ is a set of collected skills trajectories. By minimizing this term, $\phi$ integrates temporally-extended dynamics: areas connected by skills are brought closer in the representation space. This term will be referred to as the *boredom* term. Its exploratory virtue is discussed in the following.

**How does boredom help exploration ?** The interplay between the high-level policy reward function (4) and this boredom term (5) induces a progressive exploration mechanism. In effect, $\pi_{\text{hi}}$ tends to sample skills that travel further, i.e. with larger $R^{\text{hi}}$. The more often a skill is sampled, the less rewarding it becomes to $\pi_{\text{hi}}$ due to the minimization

of $\mathcal{B}(\phi)$. This will favor sampling the remaining (under-sampled) skills, hence encouraging the exploration of less visited parts of the state space. This mechanism dynamically fights what can be considered as accumulated *boredom* along over-sampled skills trajectories which increases the agent curiosity and urge it to explore.

Finally, the proposed objective to train the representation $\phi$ consists in the base objective (2) augmented with the boredom term (5), and can be written as

$$\mathcal{L}_{\text{TATC}}(\phi; \mathcal{D}_s, \mathcal{D}_{\pi_\mu}) \triangleq \mathcal{L}_{cont}(\phi; \mathcal{D}_{\pi_\mu}) + \beta'\mathcal{B}(\phi; \mathcal{D}_s) , \quad (6)$$

with $\beta'$ a hyperparameter controlling the strength of boredom term.

## 4 TEMPORAL ABSTRACTIONS AUGMENTED TEMPORALLY-CONTRASTIVE LEARNING (TATC) IN THE NON-UNIFORM PRIOR SETTING

The proposed approach consists in simultaneously training the representation $\phi$ and the hierarchical agent $(\pi_{\text{low}}, \pi_{\text{hi}})$. The agent progressively extends the explored area while maintaining the previously collected knowledge. To do so, in the non-uniform-prior setting, the agent switches with some probability $p_{rw}$ between following a uniformly random policy $\pi_\mu$ and executing the hierarchical policy (skills). The latter helps reach further areas more efficiently, where data collected by $\pi_\mu$ is used to train the representation $\phi$. Along their training, the skills progressively extend to cover newly discovered areas. Algorithm 1 provides a pseudocode of the proposed approach, in the non-uniform-prior setting.

## 5 EXPERIMENTS

In this section, we investigate the behavior of TATC in two types of environments: gridworld environments with discrete state and action spaces, and continuous navigation environments for continuous state spaces (MuJoCo, Todorov et al. [2012]).

For our method, we learn a 2D representation ($d = 2$), and define $\Omega$ as a set of 8 unit vectors equally spaced on the unit sphere. These directions are concatenated to the input state of $\pi_{\text{low}}$. Implementation details of all the experiments in this section can be found in Appendix C.

### 5.1 GRIDWORLD

We evaluate our approach in three gridworld domains: U-MAZE, T-MAZE and 4-ROOMS. These environments, visualized in Figure 2, raise different explorations challenges. U-MAZE is a simple but relevant environment to test the

---

**Algorithm 1** TATC in the non-uniform prior setting

1: **Input:** $L, c, p_{rw}, N$
2: **for** $iteration = 1, 2, \ldots$ **do**
3:     $D_{\pi_\mu} = \emptyset, D_s = \emptyset$
4:     **for** $batch = 1, 2, \ldots, N$ **do**
5:         Reset to $s_0$ with probability $p_r$.
6:         $p \sim Unif([0,1])$
7:         **if** $p < p_{rw}$ **then**
8:             Run the uniformly random policy $\pi_\mu$ to collect $L$ random walk trajectories $\{\tau_i'\}_{i=1}^L$ of $c$ steps each.
9:             $D_{\pi_\mu} \leftarrow D_{\pi_\mu} \cup \{\tau_k'\}_{k=1}^L$
10:         **else**
11:             Run $(\pi_{\text{hi}}, \pi_{\text{low}})$ to collect $L$ consecutive skills' trajectories $\{(\tau_k, \boldsymbol{\delta}_k)\}_{k=1}^L$ and their corresponding directions
12:             $D_s \leftarrow D_s \cup \{(\tau_k, \boldsymbol{\delta}_k)\}_{k=1}^L$
13:         **end if**
14:     **end for**
15:     Optimize the policies $(\pi_{\text{hi}}, \pi_{\text{low}})$ using the intrinsic objectives 3 and 4
16:     Optimize $\phi$ so as to minimize $\mathcal{L}_{\text{TATC}}(\phi; \mathcal{D}_s, \mathcal{D}_{\pi_\mu})$ (Equation 6).
17: **end for**

---

dynamics-awareness of the representations;[2] T-MAZE raises the challenge of splitting the exploration focus at an intersection while maintaining the covering in both corridors; 4-ROOMS is similar to U-MAZE[3], but requires learning more controlled skills for a useful exploration.

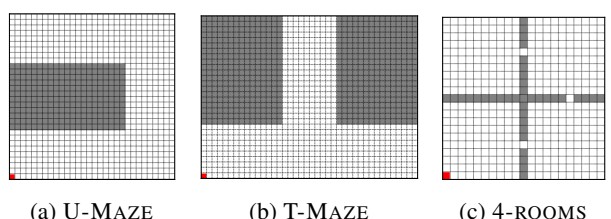

(a) U-MAZE      (b) T-MAZE      (c) 4-ROOMS

Figure 2: Gridworld domains. The fixed initial state $s_0$ is highlighted in red.

### 5.1.1 Progressive Representation Learning

Figure 3 shows the evolution of the representations throughout training. The agent progressively explores the environment starting around $s_0$, and builds the representation by continuously integrating newly discovered parts.

---

[2]The presence of the wall makes L2-distance in xy-coordinates deceptive. The L2-distance in a dynamics-aware representation space should correct for that.

[3]Note that there is no door between the first and the fourth room.

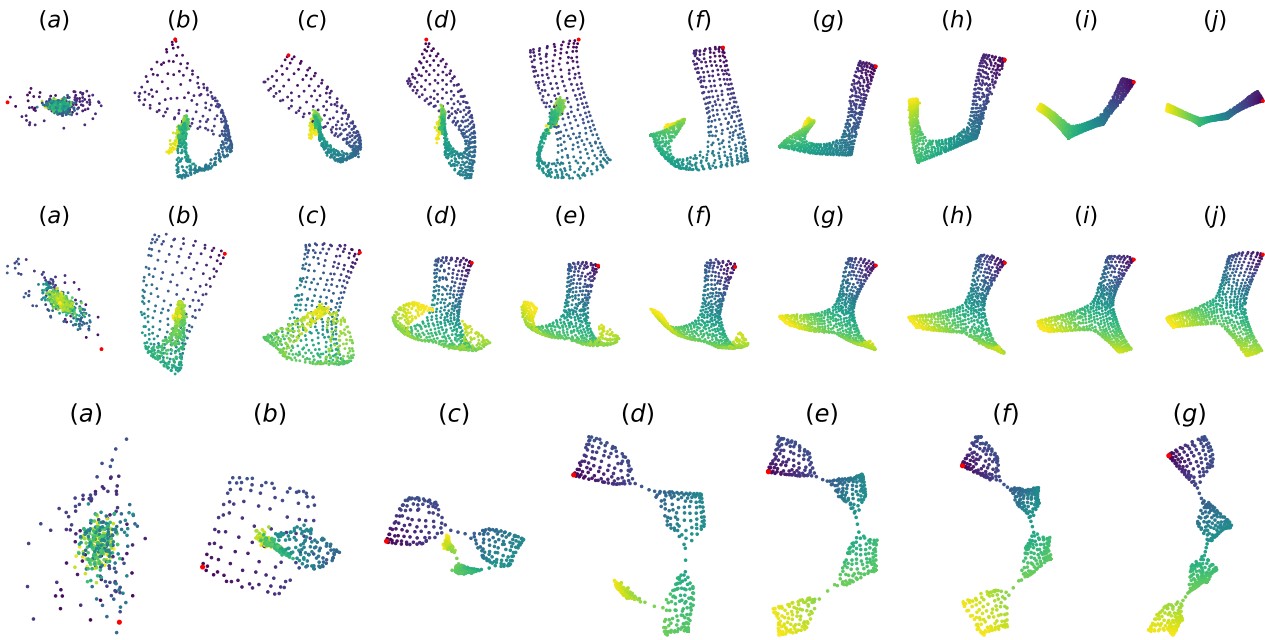

Figure 3: TATC representations learned throughout the training. For each domain, all the states are mapped with $\phi$ and represented at different stages of the training procedure. Axes scales were equalized for reliable visual appreciation. Top row: U-MAZE. Middle row: T-MAZE. Bottom row: 4-ROOMS. The colors reflect the distances in terms of the dynamics. They can be seen as quantities proportional to the length of the shortest path from $s_0$ (marked in red) to the represented state.

**U-MAZE & 4-ROOMS**. The agent starts from the bottom left corner, or room, of the maze. Figure 3 shows how the representation progressively expands away from $s_0$ along the corridor, or the rooms sequence. Note that while the agent learns to reach and represent further areas, the full domain representation *flattens*, indicating the representation's success in capturing the maze dynamics.

**T-MAZE**. The agent starts from the bottom left corner of the maze. As in the U-Maze, it starts learning to travel along the corridor until it reaches the intersection. There, the exploration focus is shared between both possible paths whose representations are progressively disentangled. Eventually, the agent fully explores both corridors and finalizes its representation. Note that, the discovery of one of the corridors did not hinder finishing the discovery of the other. The boredom term proved to be important for such property (see Appendix A).

**Importance of Boredom.** Appendix A provides an ablation study showing the importance of the boredom term for the agent's exploratory behavior and the representation's dynamics-awareness.

### 5.1.2 Evaluating the Learned Representation

We now compare our representation against LAP-REP [Wu et al., 2019] in the non-uniform-prior setting. First, to appreciate the sensitivity of LAP-REP to the uniformity of said prior, we trained LAP-REP in two settings: (i) the uniform-prior setting where the agent can be set to any arbitrary state, as done by Wu et al. [2019], and (ii) the non-uniform-prior setting defined in Section 2.3. In the following, we show that LAP-REP is sensitive to this change in distribution while TATC recovers the expressive potential that a uniform prior provides.

**Prediction.** To evaluate the learned representations, we first consider how well they linearly approximate a given task's optimal value function. To do so, we train an actor-critic agent [Mnih et al., 2016] with a linear critic on top of each representation. In Figure 4, we note a significant loss in the representational power of LAP-REP when the access to the state space is not uniformly distributed anymore. This figure also shows that TATC outperforms LAP-REP in the non-uniform-prior setting, and succeeds in recovering LAP-REP's expressive power when it is learned with the unrealistic uniform prior.

**Control.** We also compare the representations from the perspective of control, by training a deep actor-critic agent on top of each representation to solve a goal-reaching task in the same domains as above. The agent is only rewarded upon reaching the goal state ($r = 1$). Figure 5 shows that TATC consistently outperforms LAP-REP, which confirms the competitive quality of our representation.

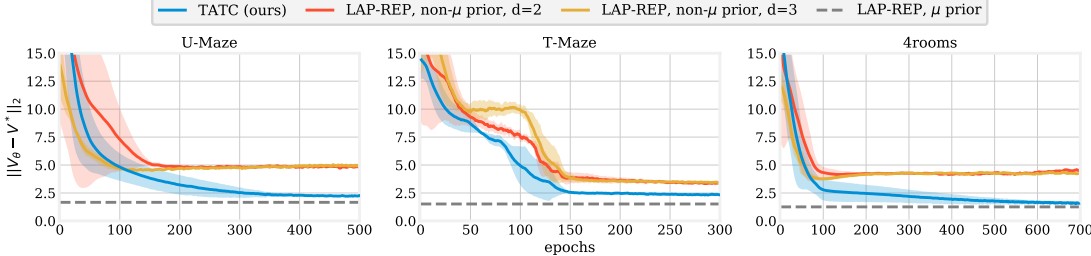

Figure 4: Learned representation's ability to approximate the value function. LAP-REP was learned in the same non-uniform-prior setting (non-$\mu$) with $d = 2$ and $d = 3$ (no improvement was observed for higher values). The dashed line gives the performance of LAP-REP in the uniform-prior setting ($\mu$). TATC outperforms LAP-REP in non-$\mu$ setting, and succeeds in recovering its expressive power when learned from the uniform prior. Performances were averaged over 5 different runs. 95% confidence intervals are shaded.

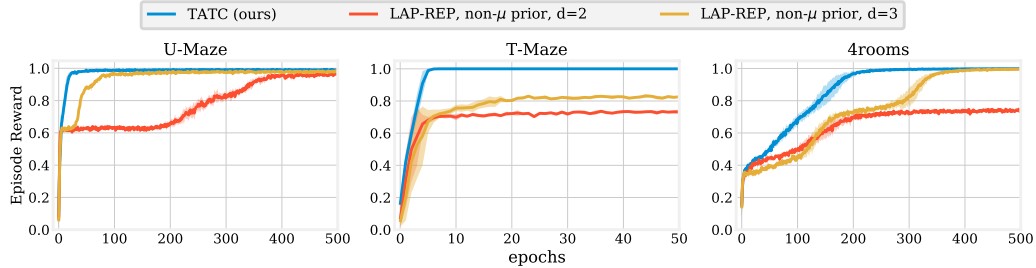

Figure 5: Control performance (Episode Reward) in the non-uniform-prior setting. Performances were averaged over 5 different runs. 95% confidence intervals are shaded.

## 5.2 CONTINUOUS CONTROL

The second set of experiments focuses on continuous state and action spaces. Here, we consider two mazes for the MuJuCo Ant agent, as shown in Figure 6. These are similar in shape to the ones from Wu et al. [2019], but are larger and thus more challenging. More details in Appendix C.

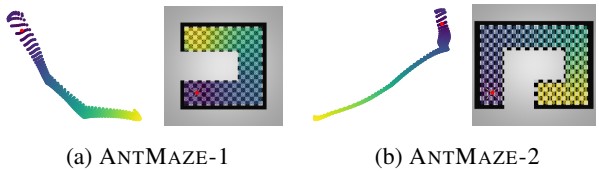

(a) ANTMAZE-1          (b) ANTMAZE-2

Figure 6: Learned TATC representations visualized on grids of positional states. Colors reflect the distance in the representation space from the initial state, highlighted in red. Axes scales were equalized. We can visually appreciate how the continuous state domains are mapped to a flatter manifold reflecting the presence of the walls.

To visualize our learned representations in these environments, Figure 6 depicts a grid of positional states in each environment domain and their mapped representations. Similarly to U-MAZE and 4-ROOMS, the learned representation translates the environment dynamics, and specifically the

presence of walls, by mapping the original state space to a flatter manifold.

### 5.2.1 Reward Shaping with Learned Representations

We demonstrate how TATC representation is able to improve an RL agent's performance when the distances in the representation space are used for reward shaping, the same setting in which Wu et al. [2019] evaluated LAP-REP. We define a goal-achieving task by setting a goal state $g$ at the end of the corridor (visualized in Appendix C). The objective is to learn to navigate to a state $s$ close enough to the goal area ($\|s - g\|_2 \leq \epsilon$). We define the reward function based on the distance in representation space (TATC and LAP-REP). More specifically, we train a soft actor-critic (SAC) agent [Haarnoja et al., 2018] to reach the goal with a **dense** reward defined as $r_t^{dense} = -\|\phi(s_{t+1}) - \phi(g)\|_2$. Similarly to Wu et al. [2019], we also compare against the half-half **mix** of the dense reward and the sparse reward $r_t^{mix} = 0.5 \cdot r_t^{dense} + 0.5 \cdot \mathbb{1}\left[\|s_{t+1} - g\|_2 \leq \epsilon\right]$.

For this evaluation, LAP-REP was learned, unlike our representation, with a *uniform* prior over $\mathcal{S}$ as in Wu et al. [2019], and $d = 2$.[4] Figure 7 shows that our representation is effec-

---

[4]Our attempts with $d = 20$ did not succeed at these reward shaping tasks.

tive in reward shaping, with both **mix** and **dense** variants, and enjoys a comparable if not superior dynamics-awareness to LAP-REP. This result stands while TATC is learned from a non-uniform prior which is less advantageous than the uniform prior used to train LAP-REP. Note that LAP-REP with a non-uniform prior fails to guide till the goal.

Finally, these results further confirm the conclusions drawn from the gridworld experiments and positions TATC as a competitive alternative to LAP-REP in this challenging setting.

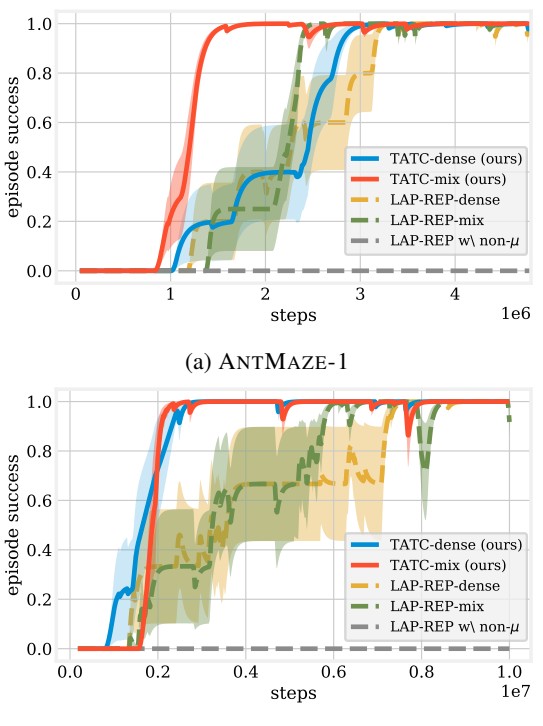

(a) ANTMAZE-1

(b) ANTMAZE-2

Figure 7: Reward shaping using learned representations: performances were averaged over 5 different runs. 95% confidence intervals are shaded. Curves were exponentially smoothed (0.9) for better visualization.

### 5.2.2 Evaluating the Learned Skills

To evaluate the exploratory potential of TATC's skills, we compare the learned skills in ANTMAZE-1 against 2 task-agnostic skill discovery methods, DIAYN [Eysenbach et al., 2019], and DCO [Jinnai et al., 2020]. DIAYN is a mutual information-based approach to learn diverse set of skills, while DCO skills are based on a temporally-contrastive representation, similarly to TATC. More specifically, DCO requires a pretrained LAP-REP which approximates the Laplacian's second eigenvector; also called the Fiedler vector. We train the required representation, as well as DCO, with the advantage of data collected from a *uniform* prior over $\mathcal{S}$. For a fair comparison, we train 8 skills for both meth-

ods (DCO and DIAYN). Once trained, the skills learned by each method are fixed and used to train a discrete high-level policy that can select across the available skills to solve a goal-reaching task with a sparse reward function $r_t = \mathbb{1}\left[\|s_{t+1} - g\|_2 \le \epsilon\right]$.

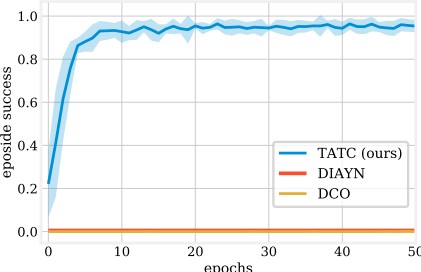

Figure 8: Skills Evaluation: performance gathered from 5 independent runs. 95% confidence intervals are shaded. Curves were exponentially smoothed (0.9) for better visualization.

The sparsity of the reward naturally poses a challenge as no additional signal can guide the agent towards the goal, unlike the evaluation setting of DCO and DIAYN by Jinnai et al. [2020]. Figure 8 shows that the skills learned by TATC quickly assist to complete the task while the skills learned with DCO and DIAYN do not. DIAYN's limited performance in difficult sparse-reward navigation tasks was also confirmed by Kamienny et al. [2021]. These results suggest that in order to succeed, DCO and DIYAN skills may require a richer signal like the distance-based dense reward used by Jinnai et al. [2020] to evaluate both of them – and where they show similar performances.

## 6 RELATED WORK

Our work is related to self-supervised learning [Bromley et al., 1993, Chopra et al., 2005], which brought recent advances in representation learning [Bachman et al., He et al., 2020, Chen et al., 2020, Grill et al., 2020, Caron et al., 2020]. These techniques have naturally been adapted to RL, especially contrastive methods. While some of these benefited from visually contrasting observations [Laskin et al., 2020, Yarats et al., 2021], others leveraged temporal contrasts to learn useful representations [Mazoure et al., 2020, Stooke et al., 2021, Li et al., 2021], which fall closer to our work.

We designed our covering policy as a hierarchical agent. This has actually been the default setting to model temporally-extended actions [Sutton et al., 1999]. Our work shares the same motivation as Vezhnevets et al. [2017] for training skills to follow latent directions. Among the large body of work on skill discovery, the eigenoptions framework [Machado et al., 2017] and its extensions [Machado et al.,

2018, 2021, Jinnai et al., 2020] are probably the closest to our skill training scheme. Eigenoptions also fit in the directional skills definition as they are trained to travel along the directions defined by the eigenvectors of the Laplacian. These vectors have a dimensionality of $|\mathcal{S}|$ which can be very large. To contrast, our directional skills are defined by an arbitrary and diverse set of directions in the learned representation space of adjustable dimensionality, which offers more tractability. TATC skills and eigenoptions also share an interesting connection to mutual information based intrinsic control methods [Hansen et al., 2020], which we discuss in Appendix E. Indeed, a similar discussion can be adapted to eigenoptions' intrinsic reward function.

The exploration mechanism in TATC emerges from the interplay between the representation and the covering policy rewarding scheme. As discussed in Section 3.4, the boredom term prevents the high-level policy from focusing exploration resources on previously over-sampled skills. This penalization induces *optimism* towards the remaining skills, in the hope of deviating from previously explored regions. Zhang et al. [2021] share a similar motivation and derive an exploration bonus that aims at directly maximizing this deviation in terms of the policy occupancy. Both methods are inspired from the *optimism-in-the-face-of-uncertainty* principle.

The incremental discovery paradigm has been previously adopted, either for exploration [Ecoffet et al., 2021], incremental skill discovery [Jinnai et al., 2020, Pong et al., 2020], or even state abstraction [Misra et al., 2020]. Finally, we use learned skills to penalize boredom [Schmidhuber, 1991, Oudeyer and Kaplan, 2009] in the representation space and encourage exploration. The idea of using skills to foster curiosity has also been investigated by Bougie and Ichise [2020].

## 7 CONCLUSION

The Laplacian representation as proposed by Wu et al. [2019] made the benefits of spectral methods affordable in large state spaces. Unfortunately, the quality of this representation is strongly tied to the uniformity of its training data distribution, as shown is Section 5. This has motivated the method proposed in this work where we reconcile a similar temporally-contrastive representation with exploration demanding settings. Our approach leverages the practical skill training framework that such representations allow. The learned skills are used to better cover the state space and hence learn a better representation. We validate our method in tabular as well as continuous environments. Our representation learned in a non-uniform-prior setting shows a comparable representational power to the one acquired by a Laplacian representation from a uniform prior, and our skills proved to be competitive in hard continuous control tasks. With these results, we hope to bring such representations'

applicability one step closer to realistic contexts.

We have proposed to augment the representation objective with temporal abstractions captured in the acquired skills. This benefits exploration by inducing a boredom-fighting mechanism, and enforces the representation's dynamics-awareness. Intuitively, this augmentation can be seen as bringing temporally close regions (connected through skills) closer in the representation space. This observation may motivate further investigations on how temporal abstractions can improve the representational potential and benefit task-agnostic representations in RL.

## ACKNOWLEDGEMENT

AE would like to thank Emmanuel Bengio, Amy Zhang, and Ahmed Touati for the insightful discussions, and Jad Kabbara and Riashat Islam for their feedback on an earlier version of this paper. We also thank all the reviewers for their feedback and valuable suggestions. The authors acknowledge the financial support of CIFAR, NSERC and Mila.

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
