# OpenReview forum: "Temporal Abstractions-Augmented Temporally Contrastive Learning: An Alternative to the Laplacian in RL"
_auai.org/UAI/2022/Conference — UAI 2022 Poster_

### Official Review · Reviewer_jJAZ · 2022-04-12

**Q2(1) Originality/Novelty:** 2
**Q2(2) Significance/Impact:** 2
**Q2(3) Correctness/Technical Quality:** 3
**Q2(6) Clarity Of Writing:** 4
**Q6 Overall Score:** 5
**Q8 Confidence In Your Score:** 4

**Q1 Summary And Contributions:**

The authors propose TATC, an approach for learning representations for reinforcement learning which is strongly inspired by Lap-Rep. TATC extends Lap-Rep by combining representation learning with learning skill-based covering policies which are in turn used to explore the state space. This helps to overcome shortcomings of Lap-Rep under the non-uniform prior assumption as demonstrated in experiments. Key contributions are the introduction of TATC and demonstrating its properties in experiments.

**Q2 Assessment Of The Paper:**

More detailed information regarding each of these aspects is given below:

**Q2(4) Quality Of Experiments (Optional):**

3: Good: The experimental evaluation is adequate, and the results convincingly support the main claims.

**Q2(5) Reproducibility:**

2: Fair: Key resources (e.g., proofs, code, data) are unavailable but key details (e.g., proof sketches, experimental setup) are sufficiently well-described for an expert to confidently reproduce the main results.

**Q3 Main Strengths:**

TATC works well in the considered experiments.

Mainly well written and easy-to-follow paper.

Hierarchical RL for exploration is sensible.

**Q4 Main Weakness:**

I didn't see the code of paper being available or the intention to make the code available upon publication. This clearly limits the reproducibility of results.

While I understand that Lap-Rep is the key baseline as the approach is developed as an extension, a comparison with other representation learning methods would help place the paper better in existing work (e.g., Augmented Temporal
Contrast, or bisimulation-based approaches).

I am missing a deeper discussion on which properties a good representation should satisfy. In particular, the authors make exploration part of the representation learning through "boredom". While this seems to work well it also entangles the exploration of the used RL algorithm and that induced by the learning of the representations.

Complexity of experiments is limited.

**Q5 Detailed Comments To The Authors:**

The interaction of exploration through properties of the representation and that of the used RL algorithm should be discussed and investigated in more detail. It appears that through the explicit "boredom" term a non-trivial entanglement takes place. In my opinion, the paper would benefit from understanding in which aspects this is orthogonal to existing exploration algorithms and in which not. I feel this is particularly important as the boredom-term has the same form as first term in the objective but this first term does not appear to be sufficient to ensure exploration.

I would be curious to see what happens if the orthonormality constraints are included. The authors say that this can make representation learning highly-stationary. What is the consequence for the conducted experiments?

Why did you choose the particular form for the second term in Equation 2? Does simply substracting the distance of the representations not work? How important is this?

I found the explanation of the high-level policy's reward confusing initially (and I am still not sure I understood it completely). In the way the reward-value is presented, it is independent of the sampled direction $\delta_k$. Is that correct? Or should this be read differently?

The experimental section is missing minimal information on the used RL agent (I know it is available in the appendix but I think at least some information must be part of the main paper).

--

I thank the authors for their comments in the rebuttal, which have made some of their arguments more explicit. I still think that certain aspects of the paper can and should be improved and if that happens in line with the provided rebuttal, I am positive about the paper. I have increased my score.

**Q7 Justification For Your Score:**

The paper is well written, proposes an intuitive approach, and seems to work well in the presented experiments. The level of originality is limited (in particular the extensions are straightforward, probably with the exception of the inclusion of the boredom term for which an extended discussion is necessary). Nevertheless, the proposed approach is interesting but for recommending acceptance I would want to see an extended comparison with other representation learning approaches for RL.

**Q9 Complying With Reviewing Instructions:**

1: Yes.

---

### Official Review · Reviewer_Cpf4 · 2022-04-14

**Q2(1) Originality/Novelty:** 3
**Q2(2) Significance/Impact:** 3
**Q2(3) Correctness/Technical Quality:** 3
**Q2(6) Clarity Of Writing:** 3
**Q6 Overall Score:** 6
**Q8 Confidence In Your Score:** 3

**Q1 Summary And Contributions:**

In this paper, the authors discuss the problem of reinforcement learning by using the Laplacian representation under the non-uniform-prior setting. Hence the authors propose the TATC framework. Technologically, the authors first improve the constraint term of Laplacian for the non-uniform prior setting. Then, the authors propose the boredom term for reinforcement learning. The authors also provide the experimental result on the GridWorld and Continuous control datasets.

**Q2 Assessment Of The Paper:**

More detailed information regarding each of these aspects is given below:

**Q2(4) Quality Of Experiments (Optional):**

3: Good: The experimental evaluation is adequate, and the results convincingly support the main claims.

**Q2(5) Reproducibility:**

3: Good: Key resources (e.g., proofs, code, data) are available and key details (e.g., proofs, experimental setup) are sufficiently well-described for competent researchers to confidently reproduce the main results.

**Q3 Main Strengths:**

1.	The authors focus on an interesting problem, how to improve the Laplacian representation in the non-uniform case.
2.	The manuscript is well-written and the experiment results are sound.


**Q4 Main Weakness:**

1.	According to the manuscript, Equation (2) is one of the most important contributions, it illustrates how the proposed method addresses difficulty for the non-uniform prior setting. But it is unclear why the authors can improve the constraint term.
2.	Moreover, it is not clear the connection between section 3.2 and 3.3. Section 3.3 introduces a hierarchical RL approach. But we can replace \phi(x) with any other feature extractor. It seems that the proposed hierarchical RL approach is not devised for Laplacian representation.
3.	It is suggested that the authors should consider more challenging tasks.


**Q5 Detailed Comments To The Authors:**

Please refer to the aforementioned Strong points and weak points.

**Q7 Justification For Your Score:**

I have read the through paper. I am confident in the assessment but not absolutely certain.

**Q9 Complying With Reviewing Instructions:**

1: Yes.

---

### Official Review · Reviewer_DSQZ · 2022-04-16

**Q2(1) Originality/Novelty:** 3
**Q2(2) Significance/Impact:** 3
**Q2(3) Correctness/Technical Quality:** 3
**Q2(6) Clarity Of Writing:** 4
**Q6 Overall Score:** 7
**Q8 Confidence In Your Score:** 3

**Q1 Summary And Contributions:**

This paper introduces TATC a temporally-contrastive method for learning a task-agnostic neural state representation.  Building from Lap-Rep, it aims to handle situations lacking uniform sampling of the state-space through augmenting representation with a learned covering strategy.

The main innovation is to augment contrastive representation-learning with multi-level policy learning to increase coverage and exploration.  They compare their approach with Lap-Rep and perform a number of ablations.

**Q2 Assessment Of The Paper:**

More detailed information regarding each of these aspects is given below:

**Q2(4) Quality Of Experiments (Optional):**

3: Good: The experimental evaluation is adequate, and the results convincingly support the main claims.

**Q2(5) Reproducibility:**

3: Good: Key resources (e.g., proofs, code, data) are available and key details (e.g., proofs, experimental setup) are sufficiently well-described for competent researchers to confidently reproduce the main results.

**Q3 Main Strengths:**

The paper is very well written, with clear motivations, claims, and technical exposition.

The high-level idea of interfacing representation with exploration is interesting and seems like a necessary component of future approaches.

The experimental results are well done, barring the caveats I have listed below.

**Q4 Main Weakness:**

It is difficult to meaningfully attribute the performance improvements over Lap-Rep to the main contributions of the paper because.  The appendix suggests that the dimensionality $d$ of the learned space was preserved between Lap-Rep and TATC, but there's no information on other possible differences that could confound the result, e.g. the number of parameters between the networks used in the different approaches.

I have some skepticism that the exploratory regime defined here will handle significantly more complex tasks, specifically, in domains where exploring requires some non-trivial amount of planning.  For instance, one can imagine a robotic environment with a door and on the other side of the door is a rich and complex environment.  Is there any reason or evidence to suggest that the exploration strategies learned here would be sufficient to carry out the motor tasks to open the door?  It would be good to see more experimental evaluation in scenarios where exploration is hard.

**Q5 Detailed Comments To The Authors:**

Equation (4) and the section around it could be expressed with a little less ambgiutiy.  It’s unclear (to me at least) whether $s_f$ is a typo and instead should be $s_k$, or if $R^\{hi}(s_k^{hi}, \delta_k)$ is somehow invariant to $k$, or there’s some implicit rebinding of variables whereby $f$ in this context really means $k$ or $k+1$.

As a consequence, I’m not certain whether the reward function favors taking large steps at each point in the sequence or only cares about the end-points.

Do you have any intuitions (or results) on the existence of degenerate solutions?  I could imagine that unconstrained, there could be hi/low policies and $\phi$ functions that are good w.r.t. Eq 3 and 4 but do not capture any meaningful notion of exploration.  For instance, $\phi$ could aim to maximally separate everything.  If true, is the contrastive loss sufficient and necessary to prevent this from occurring in practice?

**Q7 Justification For Your Score:**

The paper contributes novel ideas to representation learning in interactive environments, relaxing nontrivial assumptions of prior work.  While I have some misgivings that this approach by itself will scale to more interesting problem domains, it seems plausible that this could be an important basis for new approaches.

**Q9 Complying With Reviewing Instructions:**

1: Yes.

---

### Decision · Program_Chairs · 2022-05-15

**Decision:**

Accept (Poster)

**Comment:**

Meta Review: The paper proposes a new representation learning technique based on augmented temporally constrastive learning, which presents an alternative to the Laplacian in RL.  This is a nice contribution that advances the state of the art in representation learning for RL.  The authors are encouraged to follow the reviewers' suggestions when preparing the final version of the paper.